# Readability of Informed Consent Forms for Medical and Surgical Clinical Procedures: A Systematic Review

**DOI:** 10.3390/clinpract15020026

**Published:** 2025-01-24

**Authors:** José Manuel García-Álvarez, Alfonso García-Sánchez

**Affiliations:** 1Resident Intern of Family and Community Medicine, Hospital Comarcal del Noroeste, 30400 Caravaca, Murcia, Spain; 2Health Sciences Program, Catholic University of Murcia (UCAM), 30107 Guadalupe, Murcia, Spain; 3Medical Specialist in Anesthesiology and Critical Care, Anesthesiology and Critical Care Department, Hospital de la Vega Lorenzo Guirao, 30530 Cieza, Spain; agarcia@ucam.edu; 4Clinical Simulation Instructor, Faculty of Nursing, Catholic University of Murcia (UCAM), 30107 Guadalupe, Murcia, Spain

**Keywords:** informed consent forms, readability

## Abstract

**Background/Objectives**: The wording of informed consent forms for medical or surgical clinical procedures can be difficult to read and comprehend, making it difficult for patients to make decisions. The objective of this study was to analyze the readability of informed consent forms for medical or surgical clinical procedures. **Methods**: A systematic review was performed according to the PRISMA statement using PubMed, Embase, and Google Scholar databases. Primary studies analyzing the readability of informed consent forms using mathematical formulas published in any country or language during the last 10 years were selected. The results were synthesized according to the degree of reading difficulty to allow for the comparison of the studies. Study selection was performed independently by the reviewers to avoid the risk of selection bias. **Results**: Of the 664 studies identified, 26 studies were selected that analyzed the legibility of 13,940 forms. Of these forms, 76.3% had poor readability. Of the six languages analyzed, only English, Spanish, and Turkish had adapted readability indexes. Flesch Reading Ease was the most widely used readability index, although it would be more reliable to use language-specific indices. **Conclusions**: Most of the analyzed informed consent forms had poor readability, which made them difficult for a large percentage of patients to read and comprehend. It is necessary to modify these forms to make them easier to read and comprehend, to adapt them to each specific language, and to carry out qualitative studies to find out the real legibility of each specific population.

## 1. Introduction

The primary function of informed consent forms for medical or surgical clinical procedures is to inform patients about the procedure they are about to undergo [1,2].

In order for these informed consent forms to adequately fulfill their informative function, it is necessary that they include sufficient information in accordance with current scientific evidence [3,4]. It is also important that the written language used in these informed consent forms is adapted to the specific characteristics of the persons to whom they are addressed. This facilitates the reading and comprehension of the information they provide [5,6].

The complete and comprehensible information in these informed consent forms allows for a free, active, and voluntary decision to be made. This ensures compliance with bioethical principles and existing legal requirements regarding the informed consent process [7,8].

Readability is the quality that allows us to evaluate the ease of reading written documents using mathematical formulas that measure the length of the words and phrases they contain. It is considered that the shorter the length of words and phrases used in a written text, the easier it is to read and comprehend [9,10].

There are different formulas that make it possible to analyze the readability of informed consent forms by generating a score that will indicate the ease of reading or the academic level that would be necessary to be able to read and comprehend these forms. All the formulas consider that the higher the score or the lower the academic level required, the easier it will be to read the written text. The most commonly used formulas for analyzing the readability of informed consent forms are presented below [11,12,13,14,15,16].

The Flesch Readability Test or Flesch Reading Ease analyzes the readability of documents written in English based on the average number of words per sentence and syllables per word. This results in a score ranging from 0 (very difficult to read) to 100 (very easy to read), grouped into seven degrees of reading difficulty. It is considered that a score above 60 could be easily read by the majority of the population (Table 1) [11].

Flesch–Kincaid Level predicts reading ease based on the level of education an American person should have completed to be able to easily read and comprehend a given text. The reading ease score with this formula ranges from 0 to 18 grades. A level equal to or below eighth grade is considered adequate for a text to be easily read and comprehended by 80% of American readers (Table 1) [11].

The SMOG (Simple Measure of Gobbledygook) Readability Index assesses the readability of English texts in terms of the years of education Americans need to comprehend a given text. It is calculated by counting the number of words with three or more syllables in 10 randomly selected sentences at the beginning, middle, and end of the text. Its score is between 0 and 18, considering a value equal to or lower than 8 is adequate to be easily read by most people (Table 1) [12].

The Gunning Fog Readability Index analyzes the readability of English texts by calculating the years of formal education a person needs to comprehend the text on first reading. The score is based on the number of complex words per sentence contained in three randomly chosen 100-word passages. It is scored between 0 and 20, considering that if the score is lower than 8, the comprehension of the text is usually almost universal (Table 1) [13].

The Szigriszt Pazos Perspicuity Formula or Flesch–Szigriszt Readability Index is an adaptation of the Flesch Readability Test for texts written in Spanish. Although it also presents a score from 0 to 100 that is divided into seven degrees of reading difficulty, it considers a score above 51 to be easily understandable by the majority of the population (Table 2) [14].

The INFLESZ Readability Index, based on the Flesch–Szigriszt Readability Index, is specifically validated to analyze the readability of health texts written in Spanish. It also scored between 0 and 100, although only five degrees of reading difficulty were established. It is considered that a text with a score equal to or higher than 55 can be read and comprehended by a large percentage of patients (Table 2) [14].

The Ateşman Readability Formula is a Turkish adaptation of the Flesch Readability Test. Although it also presents a score from 0 to 100, it is grouped into five categories, and a value above 50 is considered to present an almost universal comprehension for the Turkish population (Table 3) [15].

The Bezirci–Yılmaz Readability Formula uses the total number of sentences in the text, the number of words, the number of syllables, the number of letters, and the number of words with more than four syllables to determine the readability of the text written in the Turkish language. The score obtained provides a value of the Turkish educational level needed to be able to read the text easily. It is considered that a value below 8 can be read and comprehended by the majority of the Turkish population (Table 3) [15].

It would be interesting to know the readability of informed consent documents for medical and surgical clinical procedures written in different languages to determine whether they can guarantee the ethical and legal functions they are intended to perform. Most studies on the readability of informed consent forms have been conducted with a highly educated Western population, usually excluding minority groups. Therefore, it is necessary to conduct studies that include all population groups in order to adapt these forms to the characteristics of the target population in order to facilitate their reading and enable their comprehension [5,6,7,8].

It was decided to review the clinical informed consent forms because their wording and content are less controlled than those of informed consent forms for research, being susceptible to greater variability.

The objective of this study was to determine the readability of informed consent documents for medical and surgical clinical procedures in different languages and to analyze the relationship between the different indices used.

## 2. Materials and Methods

### 2.1. Study Protocol

This review was conducted in accordance with the PRISMA 2020 (Preferred Reporting Items for Systematic Reviews and Meta-Analyses) guidelines, although it has not been registered [16].

### 2.2. Search Strategy

A literature search was conducted in PubMed, Embase, and Google Scholar databases during June 2024. The search strategy used the keywords “informed consent form”, “informed consent forms”, “informed consent document”, “informed consent documents”, and readability along with the Boolean operators AND and OR. So, the final search strategy in both databases was (“consent form” OR “consent forms” OR “consent document” OR “consent documents” OR “consent documents”) AND readability.

### 2.3. Inclusion and Exclusion Criteria

For this review, we selected primary studies that analyzed the readability of consent forms for medical or surgical clinical procedures using mathematical formulas that were published in any country or language during the last 10 years.

### 2.4. Study Selection and Data Extraction

The studies obtained after the initial search in the different databases were selected on the basis of the title and abstract. Subsequently, the initially included articles were read in their entirety to select those that fully met the inclusion criteria. For each article finally selected, the following information was collected in a table: author(s) and year of publication; country; language; scope and number of informed consent forms analyzed; formula for assessing readability used; results obtained with each formula; and the degree of difficulty of reading according to the formula used. The data in the main table were grouped into other, more specific tables to facilitate comprehension. This process was carried out independently by the authors, although those articles with discrepancies were analyzed jointly to reach a consensus decision. No other variables were included in the selected studies.

### 2.5. Data Synthesis

As the readability formulas used were different, the degree of reading difficulty (qualitative polytomous variable) was used to compare the readability of the informed consent documents analyzed in the selected studies. Therefore, a qualitative systematic review was performed to compare the studies according to the frequency and percentage of the result of this variable as a summary measure.

### 2.6. Study Risk of Bias Assessment

Given that the selected studies analyze the readability of all the medical or surgical informed consent forms available in each study setting with the different existing formulas, they present little risk of bias. Therefore, it is only necessary to control the risk of selection bias. To this end, the selection process was carried out independently by the authors, jointly analyzing those in which there was a discrepancy in order to reach a consensus decision.

## 3. Results

The literature search found 664 studies analyzing the readability of informed consent forms. A total of 35 studies remained after eliminating duplicate articles, and those that, after reading the title and abstract, were found not to use any tool to measure readability. These 35 studies were read in their entirety; eight articles were excluded because they did not use any mathematical formula to measure readability, and one article was excluded because it was a systematic review that analyzed the modification of the elements of the informed consent form and not the readability. Finally, 26 studies (Figure 1) were subjected to a detailed analysis (Table 4).

The language most frequently used in the 13,940 informed consent forms for medical and surgical clinical procedures analyzed was English, followed by Spanish and Turkish (Table 5).

The most used readability index was the Flesch Reading Ease. This index is specific to texts written in English (Table 6).

Most studies analyzed to assess the readability of informed consent forms for medical and surgical clinical procedures present difficulties reading them (76.3%) (Table 7).

The analyzed informed consents written in English, regardless of the readability index used, were mostly difficult to read (84%) (Table 8).

Most informed consent forms written in Spanish were difficult to read, regardless of the readability index used. Using indexes specific to the Spanish language (INFLESZ and Flesch–Szigriszt), it was observed that one form with reading difficulty using the Gunning Fog index, which is originally from the English language, presented a normal reading difficulty (Table 9).

A high percentage (57.9%) of the analyzed informed consent forms written in the Turkish language presented readability difficulties, especially those with English-specific indices. Turkish-specific indices (Ateşman and Bezirci–Yılmaz) obtained mostly normal readability scores for the same forms (Table 10).

The rest of the analyzed informed consent forms written in other languages presented difficult readability using specific indexes for the English language (Table 11).

## 4. Discussion

Most of the analyzed studies showed that informed consent forms for medical and surgical clinical procedures presented low readability (76.3%), regardless of the country, language, or index used to evaluate it. These results may hinder the patients’ comprehension of these forms and condition their ability to choose [17,18,19,20,21,22,23,24,25,26,27,28,29,30,31,32,33,34,35,36,37,38,39,40,41,42].

Only six languages were used in the informed consent forms analyzed. The most commonly used languages were English, Spanish, and Turkish. These results are indicative of the fact that researchers who have been more interested in knowing the actual readability of texts written in their own language have developed their own readability indices or have adapted indices originally developed for other languages [11,12,13,14,15].

It is important that patients who are going to undergo medical or surgical clinical procedures susceptible to complications are able to adequately read and comprehend informed consent forms to avoid the ethical and legal repercussions that lack of information can have [43,44,45].

Flesch Reading Ease has been the most widely used index to assess the readability of the informed consent forms analyzed, possibly because it is included in the most widely used word processing applications. It has been observed that this index only predicts 75% of the patient’s comprehension of the written documents when used in conjunction with the multiple-choice tests used to analyze the comprehension of these texts. Therefore, this index may overestimate readability and be less suitable for predicting comprehension of healthcare texts [46].

The Simple Measure of Gobbledygook (SMOG) Index was the second most used readability scale in this research. It is considered the most appropriate index for analyzing the readability of healthcare texts as it has been developed to predict 100% comprehension. It is based on more recent criteria for determining reading level and is easy to use because it only requires counting the number of words with three or more syllables and not the individual count of each syllable [46].

In most of the informed consent forms analyzed, several indices were used to assess readability. When several readability indices are used simultaneously to analyze the same text, it is advisable to choose the score of the specific readability index for that language as the most reliable. If this is not possible because there is no index for that language, the index that indicates greater reading difficulty or the need for a higher academic level for easy reading should be chosen [46].

The indices created to assess the readability of informed consent forms written in English indicate greater reading difficulty when used in other languages compared to their own indices [25,31,32,34,35]. Therefore, it is necessary to validate the indices in English in other languages or create our own scales in order to know the real readability of the texts written in each language [14,15]. Readability scores can be greatly influenced by grammatical differences between different languages [47]. Therefore, it is necessary to assess the readability of informed consent forms in Spanish-speaking patients by using specific indexes for the Spanish language (INFLESZ and Flesch–Szigriszt) in order to obtain real results [31,35].

In order to improve the readability of informed consent forms, different aspects of the text must be taken into account: content; style; presentation; format; and organization. These aspects are included in linguistic readability and topographical readability. Linguistic readability refers to the semantic and syntactic characteristics of the text. Typographic readability refers to the visual perception of the text (arrangement of the text on the page, font type and size used, use of italics or bold, etc.). Readability indices exclusively analyze linguistic readability, being necessary to perform a real evaluation of the readability of these forms by means of qualitative research techniques that also evaluate topographical readability, such as the self-report and descriptive narrative method [46,48,49,50].

To improve the linguistic and topographical readability of informed consent forms, it is necessary to take into account a series of recommendations that have proven to be useful. These recommendations include limiting the number of pages, using short and simple sentences and words, minimizing the use of abbreviations, symbols, and specific terms, avoiding sentences in passive voice, including explanatory drawings, increasing the spaces between paragraphs, and highlighting relevant information [29,34,41,48,51,52,53,54,55,56,57].

Informed consent forms should be limited to as few pages as possible and should not exceed six pages [41,55]. Words of less than three syllables, sentences of less than twelve words, and paragraphs of less than seven lines should be used [54].

It is preferable to use sans serif fonts such as Arial or Calibri with a size greater than fourteen so that they are easy for the elderly or visually impaired to read [51,55]. It is important to place the most relevant information at the beginning of each paragraph to more easily capture the attention of patients [58]. It is convenient to transform passive to active voice sentences to facilitate the identification of the subject who actually performs the action and to favor comprehension [58]. It could also be very useful to involve the specific population at whom these forms are aimed so that they participate in their drafting and in deciding on the final version [41,57,59]. Another option that could be useful to improve the readability of these forms could be for a person outside the healthcare profession but with appropriate training in biomedical language to adapt them to the language commonly used by patients by working together with the healthcare professionals involved in the informed consent process and with community education professionals [18,19,32,60,61].

In addition to the ease of reading, comprehension is also influenced by other reader factors such as academic and sociocultural level, previous knowledge of health issues, motivation, and the conditions in which reading is performed that favor or hinder concentration. Therefore, it is necessary to take into account all these factors to understand the population’s real comprehension of these informed consent forms [41,46,53,58,60,62].

Suitability is the adequacy of written texts to promote reading and comprehension in a given population. There are instruments, such as the Suitability Assessment of Materials (SAM), that analyze suitability by assessing six areas, including content, literacy demand, graphic illustrations, design and typography, learning stimulation and motivation, and cultural appropriateness. These instruments can be used to improve informed consent forms and facilitate their comprehension by patients [63,64].

The limitations of this study derive from the readability formulas used, mainly because not all long words or phrases are necessarily difficult to read, and not all short words or phrases are necessarily easy to read. In addition, these readability formulas do not evaluate typographical elements or other aspects of the text that influence the readability of these forms. Therefore, readability formulas are not able to establish with complete accuracy the ease or difficulty of reading informed consent forms for a given population.

Another limitation is the review of studies that analyze the readability of informed consent forms written in a few languages. In addition, only three languages use their own readability formulas. Therefore, the results of this review cannot be generalized to all languages and countries.

## 5. Conclusions

Most of the informed consent forms for medical or surgical clinical procedures analyzed had poor readability, making them difficult to read for a large percentage of patients.

The difficulty in reading the informed consent forms for analyzed medical or surgical clinical procedures limits their comprehension and is a major obstacle for the persons involved in making a free and voluntary decision to undergo a clinical procedure.

It is necessary to modify the informed consent forms for clinical medical or surgical procedures by establishing specific measures to facilitate reading and improve comprehension. To this end, the linguistic and sociocultural factors of the entire target population should be taken into account. This would facilitate the reading and comprehension of these forms by minority population groups that are usually excluded from studies analyzing their readability.

In order to adequately analyze the readability of informed consent forms for medical or surgical clinical procedures and obtain real results, it is necessary to use readability formulas specifically adapted for each language. In addition, qualitative studies should be carried out to analyze those aspects that are not measured by these formulas in order to determine the real readability and comprehension of these forms for a given population.

## Figures and Tables

**Figure 1 clinpract-15-00026-f001:**
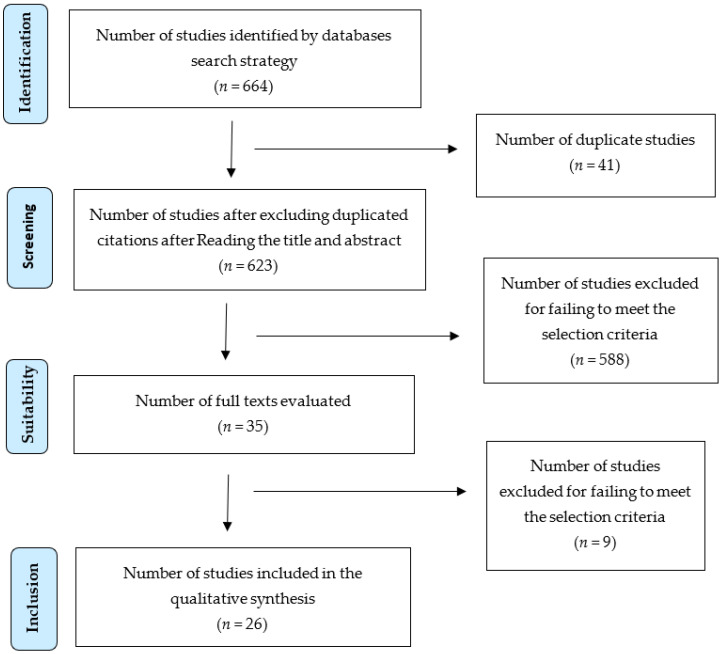
Flowchart based on PRISMA guidelines.

**Table 1 clinpract-15-00026-t001:** Indices for assessing readability in the English language.

Index	Reading Difficulty						
	Very Difficult	Difficult	Somewhat Difficult	Normal	Easy Enough	Easy	Very Easy
Flesch Reading Ease	0–30	30–50	50–60	60–70	70–80	80–90	90–100
Flesch–Kincaid level	16–18	12–16	8–12	6–8	5–6	4–5	<4
Simple Measure of Gobbledygook	17–18	13–17	8–13	7–8	6–7	5–6	<5
Gunning Fog	17–20	13–17	8–13	7–8	6–7	5–6	<5

**Table 2 clinpract-15-00026-t002:** Indices for assessing readability in the Spanish language.

Index	Reading Difficulty						
	Very Difficult	Difficult	Somewhat Difficult	Normal	Easy Enough	Easy	Very Easy
Flesch–Szigriszt	0–15	15–35	35–50	50–65	65–75	75–85	85–100
INFLESZ	0–40		40–55	55–65	65–80		80–100

**Table 3 clinpract-15-00026-t003:** Indices for assessing readability in the Turkish language.

Index	Reading Difficulty				
	Very Difficult	Somewhat Difficult	Normal	Easy Enough	Very Easy
Ateşman	1–30	30–50	50–70	70–90	90–100
Bezirci–Yılmaz	>16	13–16	8–13	4–8	<4

**Table 4 clinpract-15-00026-t004:** Details of selected studies.

Study	Country	Language	Scope	Number	Index	Result	Reading Difficulty
Lin et al., 2024 [17]	USA	English	Medical	104	FKL	11 ± 5.6 (¿?)	Somewhat difficult
García-Álvarez and García-Sánchez, 2024 [18]	Spain	Spanish	Anesthesia	4	INFLESZ	44.8 ± 1.3 (42.9–45.6)	Somewhat difficult
García-Álvarez and García-Sánchez, 2024 [19]	Spain	Spanish	Surgery	37	INFLESZ	50.7 ± 5.6 (42.5–58.7)	Somewhat difficult
Issa et al., 2024 [20]	USA	English	Spine surgery	15	FRE	69.2 ± 17.2 (10–96.4)	Normal
					SMOG	8.1 ± 2.6 (3.1–25.6)	Normal
Hannabass and Lee., 2023 [21]	USA	English	Otorhinolary-ngology	27	FRE	45.1 ± ¿?(¿?)	Difficult
					FKL	11.7 ± ¿? (¿?)	Somewhat difficult
					GF	15.5 ± ¿? (¿?)	Difficult
					SMOG	14.6 ± ¿? (¿?)	Difficult
Dağdelen and Erdemoğlu, 2023 [22]	Turkey	Turkish	Gynecology and obstetrics	18	Ateşman	38.6 ± ¿? (¿?)	Somewhat difficult
					BY	17.3 ± ¿? (¿?)	Very difficult
Morales-Valdivia et al., 2023 [23]	Spain	Spanish	Blood transfusion	45	INFLESZ	50.6 ± 4.5 (42.1–62.8)	Somewhat difficult
Peiris et al., 2022 [24]	USA	English	Radiology	399	SMOG	12.1 ± ¿? (¿?)	Somewhat difficult
					GF	10.5 ± ¿? (¿?)	Somewhat difficult
					FRE	51.6 ± ¿? (¿?)	Somewhat difficult
Dural, 2022 [25]	Turkey	Turkish	Cardiology	20	GF	16.7 ± 0.6 (¿?)	Difficult
					FRE	22.4 ± 0.8 (¿?)	Very difficult
					Ateşman	55.5 ± 3.6 (47.6–60.9)	Normal
					BY	11.6 ± 1.5 (9.2–14.5)	Normal
Meade and Dreyer, 2022 [26]	Australia	English	Orthognathic surgery	26	SMOG	12.3 ± ¿?(¿?)	Somewhat difficult
					FKL	11.5 ± ¿? (¿?)	Somewhat difficult
					FRE	52.3 ± ¿? (¿?)	Somewhat difficult
Meade and Dreyer, 2022 [27]	Australia	English	Orthognathic procedures	59	SMOG	11.2 ± ¿? (10.8–11.5)	Somewhat difficult
					FRE	40.14 ± ¿? (33.9–46.3)	Difficult
Ay and Doğan, 2021 [28]	Turkey	Turkish	Ophthalmic surgery	40	Ateşman	55.6 ± 5.7 (¿?)	Normal
					BY	10 ± 2 (¿?)	Normal
del Valle Ramírez-Durán et al., 2021 [29]	Flanders	Dutch	Medical and surgical	75	FRE	46 ± ¿? (¿?)	Difficult
Pastore et al., 2020 [30]	Australia	English	Orthopedic surgery	23	FKL	11.6 ± 1.2 (¿?)	Somewhat difficult
					SMOG	7.5 ± 0.2 (¿?)	Normal
García et al., 2020 [31]	Spain	Spanish	Cardiac surgery	38	FRE	11.9 ± 2.6 (¿?)	Somewhat difficult
					GF	16 ± 1.1 (¿?)	Difficult
					FZ	¿?	Normal
					INFLESZ	¿?	Normal
Sönmez et al., 2020 [32]	Turkey	Turkish	Medical and surgical	387	Ateşman	54.8 ± 5.7 (¿?)	Normal
					BY	9.8 ± 1.5 (¿?)	Normal
					GF	17.2 ± 1.3 (¿?)	Very difficult
					FRE	23.3 ± 1.3 (¿?)	Very difficult
Santel et al., 2019 [33]	USA	English	Radiology	399	SMOG	14.1 ± 1.2 (¿?)	Difficult
					FKL	10.5 ± 1.3 (¿?)	Somewhat difficult
					FRE	48.6 ± 5.2 (¿?)	Difficult
Sönmez et al., 2018 [34]	Turkey	Turkish	Urology	69	GF	17 ± 1.7 (¿?)	Very difficult
					FRE	23.1 ± 2 (¿?)	Very difficult
					Ateşman	55.1 ± 7.3 (¿?)	Normal
					BY	9.6 ± 1.8 (¿?)	Normal
Mariscal-Crespo et al., 2017 [35]	Spain	Spanish	Medical and surgical	11,339	INFLESZ	98.3% < 55	Somewhat difficult
					FZ	48.7 ± ¿? (¿?)	Somewhat difficult
Sivanadarajah et al., 2017 [36]	England	English	Orthopedic surgery	58	FRE	63.6 ± ¿? (61.2–66.0)	Normal
López-Picazo and Tomás, 2016 [37]	Spain	Spanish	Medical and surgical	132	INFLESZ	44.1 ± ¿? (¿?)	Somewhat difficult
López-Picazo JJ et al., 2016 [38]	Spain	Spanish	Medical and surgical	359	INFLESZ	45.8 ± ¿? (¿?)	Somewhat difficult
Nair et al., 2016 [39]	United Arab Emirates	Arabic	Medical	159	FRE	35.7 ± 3.6 (¿?)	Difficult
					FKL	12.4 ± 0.4 (¿?)	Difficult
Eltorai et al., 2015 [40]	USA	English	Surgical	11	FRE	29.9 ± 12.5 (¿?)	Very difficult
					FKL	14.9 ± 3.8 (¿?)	Difficult
					GF	17.5 ± 2.7 (¿?)	Very difficult
					SMOG	13.6 ± 2.6 (¿?)	Difficult
Vučemilo et al., 2015 [41]	Croatia	Croatian	Medical and surgical	52	SMOG	13.2 ± 1.5 (10–19)	Difficult
Boztaş et al., 2014 [42]	Turkey	Turkish	Medical and surgical	45	GF	22.9 ± ¿? (8.2–25.2)	Difficult
					FRE	20.5 ± ¿? (18.9–21.9)	Difficult
					Ateşman	33.2 ± ¿? (26.0–37.0)	Somewhat difficult

Notes: Result = mean ± standard deviation (minimum value, maximum value); ¿? = unknown values; FRE = Flesch Reading Ease; FKL = Flesch–Kincaid Level; SMOG = Simple Measure of Gobbledygook Index; GF = Gunning Fog Index; BY = Bezirci–Yılmaz Readability Formula; FZ = Flesch–Szigriszt Index.

**Table 5 clinpract-15-00026-t005:** Languages analyzed in the selected studies.

	Frequency	Percentage
English	10	38.5
Spanish	7	26.9
Turkish	6	23.1
Arabic	1	3.8
Dutch	1	3.8
Croatian	1	3.8
Total	26	100

**Table 6 clinpract-15-00026-t006:** Readability indices used in selected studies.

Index	Frequency	Percentage
Flesch Reading Ease	15	25.4
Simple Measure of Gobbledygook	9	15.3
Gunning Fog	8	13.6
INFLESZ	7	11.9
Flesch–Kincaid Level	7	11.9
Ateşman	6	10.2
Bezirci–Yılmaz	5	8.5
Flesch–Szigriszt	2	3.4
Total	59	100

**Table 7 clinpract-15-00026-t007:** Analysis of the readability of the selected studies.

Reading Difficulty	Frequency	Percentage
Normal	14	23.7
Somewhat difficult	21	35.6
Difficult	16	27.1
Very difficult	8	13.6
Total	59	100

**Table 8 clinpract-15-00026-t008:** Readability of informed consent forms written in English.

Index	Studies	Reading Difficulty (%)			
	Frequency (%)	Normal	Somewhat Difficult	Difficult	Very Difficult
Flesch Reading Ease	8 (32%)	2 (25%)	2 (25%)	3 (37.5%)	1 (12.5%)
Simple Measure of Gobbledygook	8 (32%)	2 (25%)	3 (37.5%)	3 (37.5%)	-
Flesch–Kincaid Level	6 (24%)	-	5 (83.3%)	1 (16.7%)	-
Gunning Fog	3 (12%)	-	1 (33.3%)	1 (33.3%)	1 (33.3%)
Total	25	4	11	8	2

**Table 9 clinpract-15-00026-t009:** Readability of informed consent forms written in Spanish.

Index	Studies		Reading Difficulty (%)	
	Frequency (%)	Normal	Somewhat Difficult	Difficult
INFLESZ	7 (63.6%)	1 (14.3%)	6 (85.7%)	-
Flesch–Szigriszt	2 (18.2%)	1 (50%)	1 (50%)	-
Flesch Reading Ease	1 (9.1%)	-	1 (100%)	-
Gunning Fog	1 (9.1%)	-	-	1 (100%)
Total	11	2	8	1

**Table 10 clinpract-15-00026-t010:** Readability of informed consent forms written in Turkish.

Index	Studies	Reading Difficulty (%)			
	Frequency (%)	Normal	Somewhat Difficult	Difficult	Very Difficult
Ateşman	6 (31.6%)	4 (66.7%)	2 (33.3%)		
Bezirci–Yılmaz	5 (26.3%)	4 (80%)			1 (20%)
Flesch Reading Ease	4 (21.1%)			1 (25%)	3 (75%)
Gunning Fog	4 (21.1%)			2 (50%)	2 (50%)
Total	19	8	2	3	6

**Table 11 clinpract-15-00026-t011:** Readability of selected studies written in other languages.

Index	Studies	Readability
	Frequency	Difficult
Arabic: Flesch Reading Ease	1	1
Arabic: Flesch–Kincaid Level	1	1
Dutch: Flesch Reading Ease	1	1
Croatian: Simple Measure of Gobbledygook	1	1
Total	4	4

## Data Availability

No new data were created or analyzed in this study. Data sharing is not applicable to this article.

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
