# Peer review of "Readability of Informed Consent Forms for Medical and Surgical Clinical Procedures: A Systematic Review"

_clinpract, 2025, doi:10.3390/clinpract15020026_

Round 1
Reviewer 1 Report
Comments and Suggestions for Authors
This is an interesting manuscript on an important topic.
My major overall comment relates to the somewhat surprising results of your systematic review. In particular, you concluded that there were only 17 publications over the past 10 years that met your inclusion criteria. And, excluding languages for which there was only a single publication, there were 14 manuscripts: 6 in Spanish, 5 in Turkish, and 3 in English. I admit that I am not an expert on systematic reviews, but it seems unexpected to me that (1) there are so few results from your search, and (2) that there are almost four times as many results in Spanish and Turkish than there are in English.
I just did my own quick-and-dirty Google search, and it showed, just to pick a few of the results:
Lin GT et al., JAMA Intern Med. 2024;184(2):214-216. doi:10.1001/jamainternmed.2023.6431, “Content and Readability of US Procedure Consent Forms.” It is published with a date of December 2023, so it should have met the timing criteria of your search in June 2024. And it isn’t obvious why this manuscript (which did use Flesch-Kincaid levels) would not have met your inclusion criteria.
Karimi AH et al., Journal of Surgical Research, Volume 296, April 2024, Pages 711-719, “Assessing the Readability of Clinical Trial Consent Forms for Surgical Specialties.”
Sivanadarajah N et al., Ann R Coll Surg Engl. 2017 Nov;99(8):645-649, doi: 10.1308/rcsann.2017.0188. “Informed consent and the readability of the written consent form.”
Mertz K et al., Journal of Hand Surgery Global Online, Volume 1, Issue 3, 149-153 July 2019. “The Reading Level of Surgical Consent Forms in Hand Surgery”
Morales-Valdivia E et al., Vol. 21 No. 4 (2023): Blood Transfusion 4-2023 (July-August), “A comprehensive analysis of the readability of consent forms for blood transfusion in Spain”
Maybe there are indeed technical reasons relating to your inclusion criteria that led to these articles being excluded, but even if that is true, it raises a question as to whether perhaps your inclusion criteria are too rigid. These articles seem to be squarely within the scope of the sorts of articles that should indeed fit within this type of analysis.
Second, your article seems to be limited to consent forms used for clinical care – i.e., it is excluding consent forms used in the setting of research. You should make that point more explicitly in your text. It is particularly important given that, since there are very specific requirements for written consent in the research setting, there is a large literature on the topic of the readability of research consent forms. Yet presumably that literature is outside of the scope of your systematic review.
And related to that point, it is noteworthy that quite a few of your references refer not to clinical consent, but research consent. For example, the references 26, 39 through 43 and 45 through 47 appear to be about consent in the research setting. Yet your use of these references in the text does not appear to highlight that difference. If indeed research consent is so similar to clinical consent that you are able to use references relating to the former, that raises the question why your actual inclusion criteria did not include articles dealing with research consent. In any event, you should probably discuss somewhere in the article why you made the decision to only be reviewing clinical consent, and not research consent.
More specific comments:
Lines 20-21: You refer to your having selected 17 studies “for a detailed analysis of their readability.” That seems somewhat confusing, since you aren’t actually analyzing those studies for “their readability.”
Line 61: The “of” needs to be deleted. And “grades” can also probably be deleted.
Lines 92-95: Unlike for the other indices, you do not describe how the scoring for this formula is computed.
Lines 109-110: Your manuscript doesn’t seem to do much in terms of the avowed goal of analyzing “the relationship between the different indices used.”
Lines 142-143: You say you excluded one article because it was a systematic review. Is that article your reference 52, Glaser et al.? If not, which is it? And most importantly, shouldn’t you be identifying and discussing that systematic review, and explaining why your review is not pre-empted by that prior review, or the ways in which your review adds value to what that prior review revealed?
Line 143: You says that 40 studies were subjected to a detailed analysis, and refer to Figure 1. Yet, looking at Figure 1, I do not see where the 40 studies show up. Under Suitability, you list 23 studies, and then indicate 6 that you exclude after determining they don’t meet your selection criteria. I don’t see the number 40 anywhere.
Line 248: You refer to researchers ‘in these countries.” Yet you were not talking about countries, but only languages. You should be more precise about the reference to countries – e.g., are you referring to any country where any of the languages you mentioned might be spoken?
Lines 301-302: You specifically refer to research subjects here. It is an odd reference, since in most of the rest of the paper, you seem to have confined your analysis to clinical consent. Thus, why is it appropriate to here be referring to research consent?
Author Response
Dear Reviewer. Attached is a document with responses to your comments.

Reviewer 2 Report
Comments and Suggestions for Authors
Introduction
I would have liked to see more detail about the ramifications of poor readability for research recruitment and inclusion of diverse populations. In the sentence that says: “It would be interesting to know the readability of informed consent documents for 110 medical and surgical procedures written in different languages to determine whether they 111 can guarantee the ethical and legal functions they are intended to perform,” please add the following important ramifications =
For instance, minoritized groups may have lower educational attainment and may be intimidated by complex language. In diverse nations with high immigrant populations such as the US, complex language can prevent the inclusion of patients who are not native speakers.
Furthermore, would have been good to review the history that may predispose underserved groups to be wary of research (e.g. Tuskegee as one example), leading prospective research participants to avoid studies due to fear of adverse events.
Finally, one of the main criticisms of human subject research is that it samples largely White, Western (and in the case of biological research) male participants.
Please include more detail about text artifacts that can make it difficult for patients to understand consents. For example, italicized wording is discouraged as it can be difficult for people with reading levels below 8th grade.
Results =
The authors note: “Most of the informed consent forms written in Spanish presented reading difficulties, 243 regardless of the readability index used (Table 12). With the use of specific indexes for the 244 Spanish language (INFLESZ and Flesch-Szigriszt), it was observed that a higher percent- 245 age of forms presented normal reading difficulties (Table 8).”
Please ensure the discussion specifies how this is particularly problematic given that researchers who are making efforts to include Spanish-speakers may get lower rates of participation due to the readability issues.
Discussion:
Besides “recommendations include limiting the number of pages, using short and 322 simple sentences and words, minimizing the use of abbreviations, symbols and specific 323 terms, avoiding sentences in passive voice, including explanatory drawings, increasing 324 the spaces between paragraphs and highlighting relevant information” … would be ideal to conclude about next steps researchers can take to ensure they have readable documents.
For example, could institutional review boards hire firms that can help PIs and clinicians re-write forms to have less medical jargon.
What consulting or external services can be provided to clinicians working to make sure their procedures are inclusive in terms of consent? Please add.
Conclusions please add something along the lines of= If consents are not readable, we exclude historically minoritized groups and perpetuate research and clinical practice based on White, Western and educated samples. That needs to be made more explicit throughout!
Author Response

(The authors gave the same response as above.)

Reviewer 3 Report
Comments and Suggestions for Authors
I find it unusual to have tables in the introduction
please add prisma 2020 checklist as supplementary material. All items should be reported in your review.
proportions could be presented as figure in the result section, instead of tables
references seem appropriate.
I am not sure if this manuscript adds to the existing literature in any field.
Author Response

(The authors gave the same response as above.)

Reviewer 4 Report
Comments and Suggestions for Authors
The article is within the Journal aim. It is original, very interesting, well focused and discussed.
I only mention two manuscripts regarding the overall reconstruction and comparisons on the regulation of informed consent that could further sustain in the introduction (on discretion of the authors) the issue of proper information, risks disclosure and entitled subject comprehension for shared and informed decision making.
Informed consent and health: a global analysis Thierry Vansweevelt & Nicola Glover-Thomas 2020 and Comparative study on informed consent regulation 2024 10.1016/j.jflm.2024.102674.
sincerely
Author Response

(The authors gave the same response as above.)

Round 2
Reviewer 1 Report
Comments and Suggestions for Authors
Thank you for appropriately revising your manuscript to explain some of the issues relating to your coverage of clinical consent and not research consent.
With regard to the five examples that I provided of articles that did not appear in your results, thank you for pointing out that I wrongly included one that was about research consent. But the other four appear to be about clinical consent, and some further googling on my part indicated that there should be several more similar articles that were not picked up by your analysis (including, for example, several that cited one of the articles I listed, namely the article by Sivanadarajah et al.). I am guessing that the inclusion of the word "informed" as a required search term led to the exclusion of quite a few on-topic articles. And most of these were about consent forms written in English.
While I appreciate that people can reasonably differ about the search terms to be used in a systematic review, it does seem to me that your review is giving a biased view of the literature when you end up with 11 articles that are looking at the analysis of consent forms written in Spanish or Turkish, and only three articles that are about the analysis of consent forms written in English, if indeed there are more than six other articles about the analysis of consent forms written in English that were not included due to how you constructed your search terms. Why not revise your terms to not require the word "informed," and see if that might significantly improve your conclusions?
Author Response
Dear reviewer.
Thank you very much for your recommendations and suggestions. They have guided us to improve the quality of the manuscript and we have tried to follow them with the utmost rigor.
Point-by-Point response to Reviewer 1
Comments and Suggestions for Authors
Thank you for appropriately revising your manuscript to explain some of the issues relating to your coverage of clinical consent and not research consent.
With regard to the five examples that I provided of articles that did not appear in your results, thank you for pointing out that I wrongly included one that was about research consent. But the other four appear to be about clinical consent, and some further googling on my part indicated that there should be several more similar articles that were not picked up by your analysis (including, for example, several that cited one of the articles I listed, namely the article by Sivanadarajah et al.). I am guessing that the inclusion of the word "informed" as a required search term led to the exclusion of quite a few on-topic articles. And most of these were about consent forms written in English.
While I appreciate that people can reasonably differ about the search terms to be used in a systematic review, it does seem to me that your review is giving a biased view of the literature when you end up with 11 articles that are looking at the analysis of consent forms written in Spanish or Turkish, and only three articles that are about the analysis of consent forms written in English, if indeed there are more than six other articles about the analysis of consent forms written in English that were not included due to how you constructed your search terms. Why not revise your terms to not require the word "informed," and see if that might significantly improve your conclusions?
Thank you very much for your comment. We have proceeded to perform a new search according to your suggestions. We have found more articles in English, although the conclusions have not been modified.
Reviewer 3 Report
Comments and Suggestions for Authors
The authors have improved the manuscript
Author Response
Dear reviewer.
Thank you very much for your recommendations and suggestions. They have guided us to improve the quality of the manuscript and we have tried to follow them with the utmost rigor.
Point-by-Point response to Reviewer 3
Comments and Suggestions for Authors
The authors have improved the manuscript
Thank you for your comment. In the latest revision the manuscript has been further improved.